# Performance Measurement Model for Sustainability Assessment of the Swine Supply Chain

**Silvana Dalmutt Kruger** [1,*] **, Antonio Zanin** [1] **, Orlando Durán** [2] **and Paulo Afonso** [3]

1 Department of Accounting, Federal University of Mato Grosso do Sul, Nova Andradina 79750-000, MS, Brazil
2 Escuela de Ingeniería Mecánica, Pontificia Universidad Católica de Valparaíso, Valparaíso 2430120, Chile
3 Department of Production and Systems, Algoritmi Research Center, University of Minho, 4804-533 Guimarães, Portugal
* Correspondence: silvana.d@ufms.br; Tel.: +55-49984197031

**Abstract:** In this paper, a model and a set of indicators for evaluating the sustainability in swine supply chains are presented and discussed. Using the Delphi method, environmental, social and economic indicators were identified (namely environmental performance indicators to evaluate soil, water, air, energy and environmental practices; social performance indicators related to human capital and social interaction; and economic performance indicators that address labor remuneration and return on investment). Subsequently, the proposed sustainability assessment model was applied for validation purposes in three different companies belonging to the swine supply chain in the southern region of Brazil. This study differs from previous ones by focusing on the sustainability assessment of the upstream and downstream of the supply chain, which are responsible for significant impacts. The performance of the studied companies from the three triple bottom-line (TBL) dimensions is significantly different. Both negative and positive impacts were found to be related to most of the specific metrics. The economic dimension presented a better performance than the environmental and social ones. Furthermore, there was a predominance of negative impacts in the environmental dimension, in relation to the soil, water, air, and energy indicators. Thus, specific actions, strategies, and policies must be designed for the different companies towards an effective and comprehensive sustainability throughout the swine supply chain. The proposed model can be extended to other companies in the same supply chain, replicated in other livestock and agribusiness industries and supply chains (such as cattle and poultry), and it can be used in different locations and including additional indicators and metrics.

**Keywords:** performance measurement; sustainability indicators; triple bottom-line; swine supply chain

## 1. Introduction

Swine supply chains generate significant environmental impacts which must be measured and correctly managed [1]. Focal companies have been leading the transformation in these supply chains, but nevertheless all companies from both the upstream and downstream, must contribute to the sustainability of the whole supply chain. In this context, sustainability indicators are important for continuous improvement processes focused on sustainable production goals [2]. Furthermore, a wide range of impacts should be assessed (i.e., economic, social, and environmental, among others).

Both positive and negative impacts can be identified throughout the life cycle of the product and the supply chain [3]. Particularly, environmental impacts resulting from production and consumption of resources in the swine supply chain affect households, firms, and society [4]. Those impacts are sensed at various levels and produce intergenerational long-lasting effects [5], which can go beyond the limits of industries and national economies [6]. The control and measurement of such impacts should consider the balance of the relationship between production, consumption, and the environment [7]. In

this sense, it should be highlighted that the relevance of evaluating production practices, through sustainability indicators, points towards methods of enabling the correction of negative impacts, observing environmental, social, and economic contexts.

The extant literature and current research have been offering some relevant contributions, both theoretical and practical, on these topics (namely agri-food [8], pig farming [9], and circular economy concepts in pig farming [10]). Also, new concepts, approaches, methods, and applications based on conceptual frameworks and empirical findings have been proposed in previous contributions. Indeed, sustainability perspectives in the supply chain have been approached using very different tools (e.g., multicriteria methods, sensitivity and risk analysis approaches), and highlighting buyer–supplier and interorganizational relationships from very different perspectives (e.g., in the automotive industry [11], the energy sector [12], considering the related externalities [13], and in specific supply chains, such as in the swine industry [14–16]). The literature offers several examples of theoretical and applied research for quantifying and establishing parameters for sustainability assessment in supply chains [17].

Sustainability indicators have been used to measure and motivate continuous improvement towards sustainable production goals [18]. In this sense, the definition of sustainability indicators for supply chains involves making choices concerning both what and how to measure [19]. In the sustainability assessment of production systems and supply chains, the environmental component has prevailed [20]. However, the isolated analysis of environmental aspects hinders a complete and long-term assessment. Indeed, the chosen metrics must allow one to implement an integrated measurement of environmental, social, and economic components [21].

Sustainability asks for the conscious and fair use of natural resources, seeking not to harm a set of aspects for future generations [22] but contributing to the continuity, durability, and perpetuity of such resources [23]. Concerns for future generations and the objectives of sustainable development require organizations to measure and demonstrate the practices aimed at the goals of the 2030 Agenda for sustainable development [24]. Furthermore, there is a challenge regarding evidencing and reporting economic, environmental, and social performance, which comprises the tripod of sustainability [25].

Agricultural production and related supply chains are interconnected with the Sustainable Development Goals (SDG) [25]. Therefore, assessing the sustainability of these supply chains becomes important within the context of the continuous improvement of production practices and initiatives towards the reduction of environmental impacts related to livestock [24]. The development of agricultural activities increasingly relies on non-renewable natural resources, and their expansion can cause negative impacts that should be minimized [12].

According to the triple bottom line approach and the Bellagio Principles [26], economic, social and environmental metrics are fundamental in sustainability assessments [21]. Specific studies about soil analysis, energy production, including greenhouse gases and their waste [27], or the impacts of waste on the environment [3], reveal the need for integrated measures to assess the sustainability of pig production [28], considering environmental, economic, and social variables in order to improve sustainability in supply chains [29], and, particularly, in livestock supply chains [20,25].

The reporting of negative and positive impacts is also important for achieving a wider sustainable development in the long term according to the authors of [30]. Indeed, sustainable development asks for positive impacts, but these impacts can go beyond a certain "expected" level or can be more focused on economic, social, or environmental dimensions. Eventually, important trade-offs and a poor development of some of the TBL's dimensions can result in negative impacts. Thus, both positive and negative impacts must be measured and analyzed.

The model proposed here can be applied throughout the livestock supply chain and, particularly, in activities developed by small producers and farmers has highlighted in [31]. Although other previous studies have identified performance indicators in swine

production (see, for instance, [17]), they have not highlighted sustainability metrics, which must include both positive and negative impacts.

Thus, in this paper, a model and a set of indicators for evaluating the sustainability of swine production are presented and discussed. The proposed model aggregates the three pillars of sustainability (environmental, economic, and social); highlights negative and positive impacts of swine production; and reflects the status quo of each farm, which allows for continuous evaluation of swine production. The model can be adapted to other realities or study environments.

Using the Delphi method with the participation of experts, environmental, social, and economic indicators were identified and used to design a sustainability assessment model which was applied for validation purposes in three different companies of the swine supply chain in the southern region of Brazil. The states of southern Brazil are known as the largest producers in the country, and the Santa Catarina state is the largest producer and exporter of pork in Brazil.

The paper is divided into six sections. The next section presents the authors' theoretical framework, highlighting the context of sustainability indicators and their relevance in the context of supply chains. The third section presents the research methodology and the proposed model. The case study is presented in the fourth section and the main results, insights, and contributions are presented and discussed in the fifth section. The last section presents the conclusions, limitations of this study, and the main opportunities for further research.

## 2. Conceptual Framework

### 2.1. Sustainability Indicators

Several international organizations committed to sustainable development, led by the United Nations (UN) Commission on Sustainable Development (CSD), have developed indicators for the assessment of sustainable development [32]. Those initiative brought together national governments, academic institutions, non-governmental organizations and experts from around the world. In 1996, the CSD published a document known as the Blue Book, which presented a set of 134 indicators.

Regarding the concern of quantifying and establishing parameters to measure and evaluate sustainability indicators, several initiatives have been developed (for example, the Bellagio Principles, from the International Institute for Sustainable Development (IISD, 1996) [26]).

In 2015, continuing the process of building the Millennium Development Goals (Agenda 21 and Agenda 2015), the United Nations Summit on Sustainable Development issued the 2030 Agenda [33]. The 2030 Agenda presents seventeen Sustainable Development Goals (SDG), composed of 169 goals and 232 indicators. Such indicators are devoted to guiding the efforts towards economic growth, in a context of the well-being of societies, nature conservation, minimization of climate change, reduction of economic inequality, and the promotion of peace and justice [34]. The indicators should provide evidence concerning past and present performance and, consequently, the aspects that should be improved. Sustainable development indicators (SDIs) must make a substantial contribution to sustainable development, and must not be used simply as a reporting tool, but as a planning instrument to shed light on strategic decisions on sustainable performance in both a broad sense and throughout particular organizations [35]. The use of smart technologies in swine farming can increase productivity and reduce negative impacts [16].

Thus, it will be possible to guide the changing processes [2] through sustainable development milestones. That assessments must help to establish constraints to long-term human activities with the perspective of balancing economic, environmental, and social dimensions [36] and to reveal the processes that contribute or do not contribute to sustainability [34]. Furthermore, with an increasing global awareness, the development of sustainable business has become an unavoidable objective for most organizations [37].

Such sustainability goals are relevant both in companies [38], as well as in the context of the whole supply chain [39].

Therefore, none of the dimensions of the triple bottom line (TBL) should be neglected [20]. The economic dimension refers to income, cost, and result indicators, while the environmental dimension refers to indicators related to natural resources [4]. The social dimension covers indicators related to human and social aspects, such as working conditions, community insertion, health and safety, human rights, etc. [40].

The development of sustainable production practices requires measures related to (i) energy and materials; (ii) environment and natural resources; (iii) social justice and community development; (iv) economic performance; (v) workers and families; and (vi) products and services [41].

Agribusinesses and the food industry are significant players in the promotion of sustainable development and constitute an important end-user of resources (in addition to being major waste generators). In this sense, the sustainability indicators of the food supply chain cover: (1) environmental dimension (energy, water, soil, transport, greenhouse emissions, climate change, waste, etc.); (2) social dimension (nutrition and health; food safety, ethics, gender equality, workforce, etc.); and (3) economic dimension (innovation, remuneration, financial return, etc.) [20].

### 2.2. Livestock Supply Chains

The negative impacts of most livestock industries [39], given their widely distributed geographic characteristics and the large number of competitors, may compromise sustainable development [42]. Furthermore, in the current economic context, which is increasingly globalized and competitive, organizations are part of an ecosystem which is characterized by networks of inter-organizational relationships. In this context, the search for competitive advantages begins at the supply chain level [36,43]. In fact, no company can exist separately; they must necessarily be part of global scale networks [40]. The analysis of a supply chain should include, in addition to suppliers and customers, transportation, and other service providers [44].

Livestock supply chains contribute directly to the following sustainable development goals (SDG): SDG 2-Zero hunger and sustainable agriculture; SDG 6-Clean water and sanitation; SDG 7-Accessible and clean energy; SDG 8-Dignified employment and economic growth; SDG12-Responsible consumption and production; SDG13-Combating climate change; SDG14-Life under water; and SDG15-Life on Earth [33].

Companies contribute significantly to jobs and income generation, particularly in rural locations which are generally economically and socially more depressed than urban regions [45]. The importance of the analysis and evaluation of the sustainability of activities developed in rural areas is also justified by regarding sustainable development [46], and especially regarding livestock production, animal welfare, and sustainability practices [17].

Livestock supply chains are positioned among the most advanced production chains in the world [28]. They adopt sophisticated production and process control technologies with high standards of efficiency and quality. For example, poultry and swine industries use intensive production systems, with animals being raised in confinement [28] according to strict sanitary controls and attending a set of international animal welfare requirements [47].

Therefore, it is necessary to make the swine supply chain much more sustainable. In addition, farmers need to receive a fair price for their products and services. In this sense, the trade-offs between environmental, economic, and social sustainability should be considered to evaluate the sustainability of swine production [3]. Previous studies developed measurements to evaluate sustainability in different contexts [19]. The indicators represent the information used to gauge and motivate progress towards the sustainable production goals set by the aforementioned international standards [41]. It is important to collect and use specific data from swine production farms [38].

Previous research reveals concerns about the increasing of swine production and the negative impacts of waste generation, including greenhouse gases and waste [1]. Due to the

benefits of mitigating greenhouse gas emissions and improving fertilization of agricultural land, manure treatment has attracted much attention in recent years [10]. Empirical research also highlighted the significant differences that exist in terms of greenhouse gas emissions from swine production between developing countries (including China) and developed countries [14].

Good management practices and the use of specific equipment and advanced technologies such as biodigesters can minimize the negative impacts of swine production [17]. Public investments and policies are also important. In Taiwan, for example, public investments in biodigesters have provided for biomass energy production, the reduction of greenhouse gases, the improvement of environmental quality, and the promotion of organic agriculture [15].

It can be highlighted that the effective transformation of waste from swine production into energy represents an opportunity to mitigate the negative impacts of these production systems. If the waste and effluents are adequately treated, it is also possible to reduce pollution in the livestock supply chain, providing an important advance for sustainable development. Small-scale intensive pig production systems represent more than half of the total number of pig farms in China, and the introduction of new feeding technologies and the upgrading of the waste management system are necessary to mitigate environmental impacts and achieve sustainability [14].

In Brazil, swine production has a significant economic and social relevance, especially regarding the generation of jobs and income. However, there is a lack of environmental management and evidence of the damage caused to the environment by swine production systems [10]. The negative impacts on the environment burden public authorities, as they increase the costs with public health and water treatment systems, and generate impacts on the quality of the air and soil [9]. We can identify several performance measurements from previous studies [4], which have been focused on sustainability indicators in farms of the swine supply chain in the southern region of Brazil and, particularly, in the state of Santa Catarina [26].

## 3. Materials and Methods

Initially, a set of sustainability indicators were identified from an extensive literature review, particularly on the sustainability of swine production, and considering the Global Reporting Initiative (GRI) model. Following this, a Delphi study was performed to get information from of experts on the most relevant sustainability indicators of the swine supply chain. The set of experts interviewed comprised 19 technicians, 2, managers and 3 academics, all of them with extensive experience in these topics. The Delphi method is usually applied to 15–20 experts who participate in 2 or 3 rounds of questions, ultimately seeking the convergence of the answers of these experts.

The use of the Delphi approach is recommended in situations in which a certain observed problem can benefit from subjective judgments, and where the individuals or specialists in question do not have a history of communicating. In this sense, the heterogeneity of the participants should be preserved (specialists with different backgrounds and experiences), aiming to assure the validity of the results. The Delphi method presents three basic characteristics: (i) it involves repeated questioning of individuals/experts; (ii) it avoids the direct confrontation of experts, but does so in an indirect, controlled, and anonymous manner by means of the method; (iii) it enables the awareness of expert opinions [27]. The research stages were the following ones.

(1) Indicators identification: in order to elaborate a structured interview, a set of sustainability performance measures were identified from an extensive literature review and the Global Reporting Initiative (GRI) model.

(2) Delphi method (1st round): 117 performance measures selected from the literature were presented through a structured questionnaire to a group of experts. Twenty-four specialists were selected based on their previous knowledge on the research topic

(e.g., Brazilian Agricultural Research Corporation). The interviews took an average time of 44 min, summing 17.60 h.

(3) Analysis of experts' opinions: the indicators that had consensus among the 24 experts were identified and a revised set of indicators was created.

(4) Delphi method (2nd round): a new questionnaire was applied to the experts by means of electronic submission to evaluate the revised set of indicators. Fifteen experts gave feedback in the second round.

(5) Final definition of the sustainability performance measures: definition of the model and the final set of 44 metrics (Figure 1 and Tables A1–A3 in Appendix A).

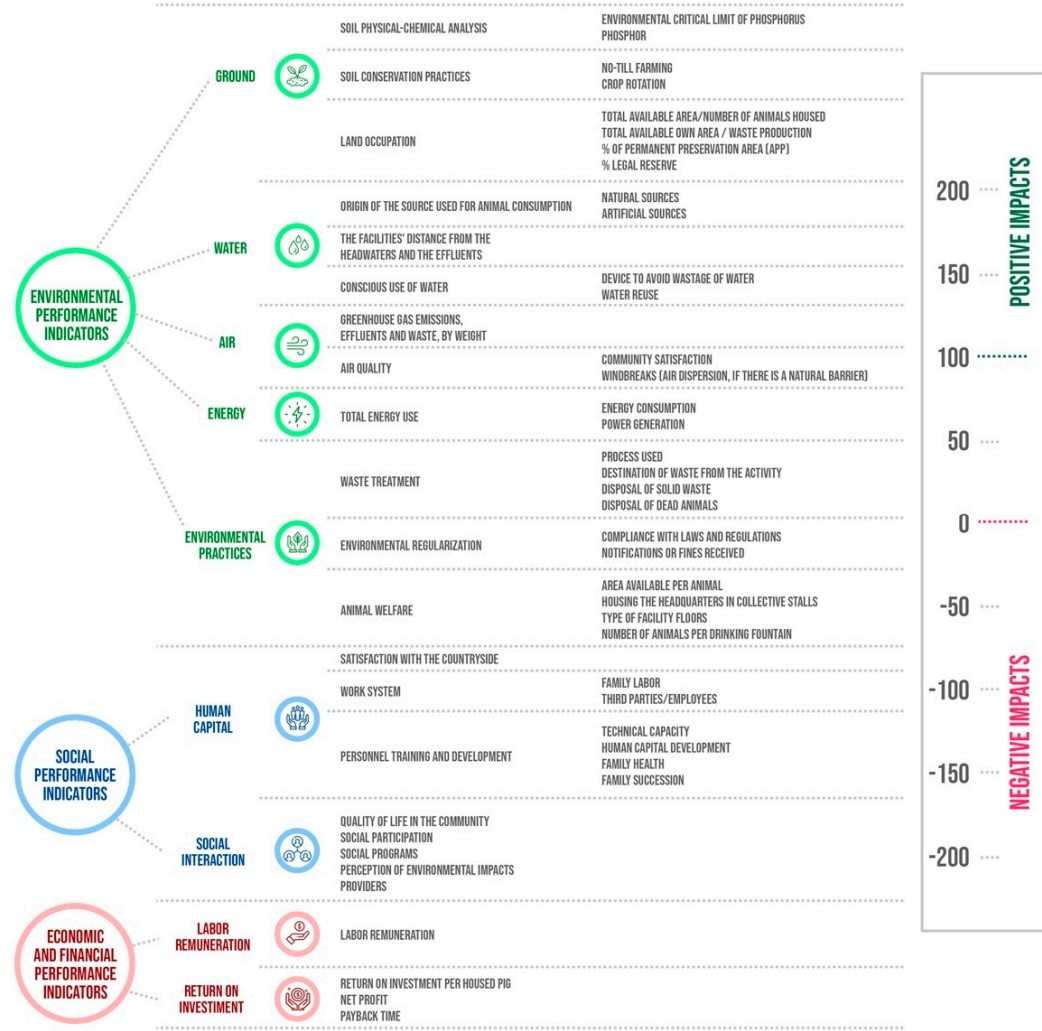

**Figure 1.** Model for sustainability assessment of the swine supply chain.

(6) Application to three farms: the sustainability assessment model was applied in three companies for validation purposes.

An indicator represents a performance measure, understood in this study as the quantification and representation of conditions related to environmental, social, and economic-financial aspects. A metric is an indicator with the following characteristics: (i) it represents a performance measure; (ii) it is related to a performance standard; (iii) it highlights negative or positive impacts which can be below or above the performance standard or target.

The sustainability assessment model was particularly designed for the swine production chain, considering its significant social and environmental impact among livestock supply chains. For validation purposes, it was applied in three farms in the upstream of a swine supply chain located in the Southern region of Brazil. The following aspects were con-

sidered: physical infrastructure, investments made, production practices, labor, electricity and water consumed, operation and maintenance costs, waste disposal facilities, etc.

The Delphi study and the field work took 90 days and 60 days, respectively, comprising the identification of the initial set of indicators, the adjustment of the model, the application to the farms, and the subsequent analysis.

Figure 1 presents the developed model composed of 44 metrics, explained in Tables A1–A3 in Appendix A.

The model follows a tree-like structure composed by the three TBL dimensions, at the top level (i.e., environmental, social and economic performance), nine constructs (e.g., five in the environmental dimension), 24 indicators and 44 metrics (the manifested variables). The environmental constructs are: (i) ground, (ii) water, (iii) air, (iv) energy, and (v) environmental practices. Furthermore, the social constructs are: (i) human capital, and (ii) social interaction. Finally, the economic-financial indicators are related to productivity and profitability through two constructs: (i) return on investment and (ii) labor remuneration.

Environmental and social indicators are relevant for all stakeholders in the swine SC. However, negative impacts are not generally measured or recognized, such as the destination of waste, the conscious use of water, the capacity of the soil to receive waste, the quality of the air, etc. In this sense, the assessment and disclosure of the negative impacts of pig production, aiming at reducing the environmental impacts of such production and maximizing any positive impacts, are extremely needed and they also play an important role within the proposed model. In the social context, it must be also demonstrated the relevance of resources management and the support given to producers, particularly in low-density rural areas, aiming at a better human capital management and social interaction. The financial indicators are related to labor remuneration and the return on the invested capital.

The metrics were defined considering the following:

1.  Measurement units: obtained from qualitative or quantitative answers, expressed in monetary values (Brazilian currency), percentages, factors or scales, dimensions, etc.;
2.  Lower and upper bounds: defined by the expert panel as presented in Tables A1–A3 of Appendix A.
3.  For all metrics, lower and upper bounds were normalized and converted to 0 and 100 points, respectively; below zero represent significant negative impacts; above 100 points are considered significant positive impacts; it was established a ceiling of $-200$ and 200 points to limit the range of values in the model.
4.  All metrics, indicators, constructs, and TBL's dimensions have the same weight given by $1/n$, where n is the number of metrics by indicator, indicators by construct, and constructs by dimension.

## 4. Case Study

According to the Annual Report published by the Brazilian Association of Animal Protein, in (2021), the exports of 750,000 tons of pork represented a revenue of more than 1.5 billion dollars for the Brazilian industry. This industry stands out as one of the main Brazilian agribusiness industries. The southern region of Brazil holds 66.5% of the Brazilian pig production according to the Brazilian Institute of Geography and Statistics (2022). This is mostly a result of the work of several small farms. The Santa Catarina state is the largest producer and exporter of pork in Brazil and the only Brazilian state authorized to export pork to Canada due to the sanitary conditions of its herds (having been considered as an area free of foot-and-mouth disease without vaccination). In 2021, the state of Santa Catarina exported 578.5 thousand tons of pork to 67 countries and international sales grew 19% compared to 2020, generating revenues of 1.4 billion dollars, according to the Integrated Agricultural Development Company of Santa Catarina (2022).

The Brazilian Southern region has been a pioneer in pig production and through research and development, has been able to create cutting-edge technologies. Moreover, this region is currently one of the largest meat producers in the world [1]. The production is

developed within a production chain in which there is a verticalized focal company, which maintains control over the other actors in the chain. They are mainly organized in the form of cooperatives of producers or agribusinesses that operate in the international market.

Indeed, most swine production in Brazil occurs in farms that have partnership agreements with large companies or cooperatives. There are about 47 cooperatives in the agribusiness industry in the State of Santa Catarina, which have a relevant contribution in generating income and jobs, as well as strengthening the development of small farms and encouraging partnership in swine production years [48].

In the studied swine supply chain, the farms receive the animals and technical assistance and feeding from focal companies. Also, the logistics among the various farms and companies is managed by the focal company. The farms are remunerated for the services provided to the focal company and receive a remuneration per animal. In this case, farms face a lower risk, since most of the costs are afforded by the focal company. Each player in the production chain receives a remuneration provided by the focal company that rules over and coordinates the entire supply chain.

Figure 2 presents the different stages and companies involved in the swine production process, which are mainly related to producers (e.g., breeding, rearing, and fattening of animals).

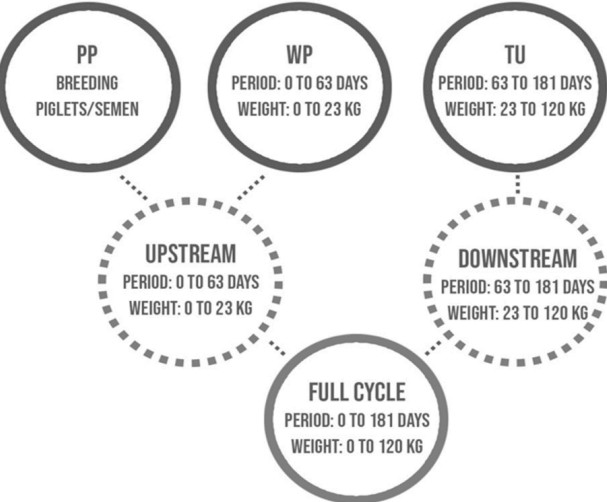

**Figure 2.** Stages of the swine production process.

The swine production unit (PP) produces sows, breeding females and semen for artificial insemination. Such piglets remain in those units until the age of approximately 28 days. The pigs are then transferred to another farm, the 'Wean to Finish' (WP), where they remain there until they are approximately 63 days old. Finally, they go to the so-called Termination Units (TU), where the process ends. In the TU the pigs reach approximately 120 kg (181 days) before being sent to the slaughter houses. In some cases, some farms run all activities (i.e., PP, WP and TU), and thus control the full cycle.

In the integrated and cooperative production model, farms invest in physical facilities and provide labor, electricity, and other maintenance costs. They are also responsible for disposing the waste produced by the production activities. They are capable of generating energy through biodigesters and using organic fertilizer for the farm's crops.

The independent producers who develop the entire process in their farms (named here full cycle) do not have partnership ties with the focal companies, bearing the whole cost of the processes. This makes them able to have commercial relationships with more companies. On the other hand, such independent producers face greater risks, since sales are strongly related to market supply and demand.

The three farms considered for the case study were selected in order to cover the three main stages of the SC (i.e., PP, WP, and TU). The principal features of each one of such farms are commented below.

Farm PP: a Piglet Production Unit of 750 animals, working independently without any partnership with a cooperative or agroindustry company. It ensures all needed sheltering and feeding for the livestock. The farm has conditions for the implementation of a waste treatment system (biodigester system).

Farm WP: responsible for the breeding of the animals, transferred to other farms when they reach 23 days (on average). Usually, 380 piglets are delivered every 23 days. This process is developed in a partnership with an agroindustry partner.

Farm TU: ensures the raising of the pigs for an average period of 115 days in the piglet finishing phase. It has a herd of 550 animals. It has a partnership with a cooperative which provides the animals and the feed required. The cooperative provides inputs to the members at their production cost (heating, labour, etc.). Administrative, technical, and operational expenses are prorated among the total number of animals produced by that cooperative member.

## 5. Analysis and Discussion of Results

As was previously mentioned, the proposed model was applied to three farms characterized by different production processes in different stages of the swine supply chain (farm PP, farm WT, farm TU). Figure 3 shows the results of the application of the model, aiming to describe and validate the model's applicability in the context of the swine production environment.

Farm PP, in the upstream of the supply chain, presented the best relative performance in environmental aspects (ground, water, air, energy, environmental practices). The superior performance can be explained, among other reasons, by the use of biodigester systems for the treatment of the swine production waste which contribute for the generation of energy and the improvement of air quality. This is similar to the findings of studies conducted in Taiwan and China [14,15], showing the impact of the use of biodigesters.

Farm PP has also a water collection and treatment system, minimizing negative impacts of its production activities. Production practices, such as crop rotation and the availability of an area to dispose of the residues generated by the production, also contribute for the positive impacts. On the other hand, both WP and TU are characterized by significant negative performance indicators in environmental aspects. This can be explained by a higher focus on the economic and financial aspects and some limitations of these companies in terms of social and environmental aspects. For example, the area available per housed animal in WP is smaller than the minimum standard, which significantly reduces animal welfare.

Social performance evaluation highlights social interaction and human capital, such as participation in the community, in class unions, and in programs focused on quality of life in rural areas, relationship with suppliers, training, workers health, business succession, and satisfaction with the rural environment. Farms PP and WP showed better results in these aspects than TU. The downstream activities in the swine supply chain seems to be more focused on economic and financial goals than environmental and social aspects. The lower social performance of the studied TU farm is related to problems with business succession and less social interaction, among other. These aspects can have a significant impact in these companies.

Regarding the economic performance indicators, it can be observed that WP and TU farms show better results in terms of both labor and capital. The high investments made by the PP farm (in machinery, equipment, air conditioning, and biodigester) to perform better from an environmental perspective may explain reduced cash flows when compared to WP and TU farms.

These trade-offs between environmental and economic dimensions of the TBL approach and between upstream and downstream activities can be important to understand

the dynamics of the swine supply chain sustainability [9], and how to deal with the challenges that the industry face towards sustainability [13].

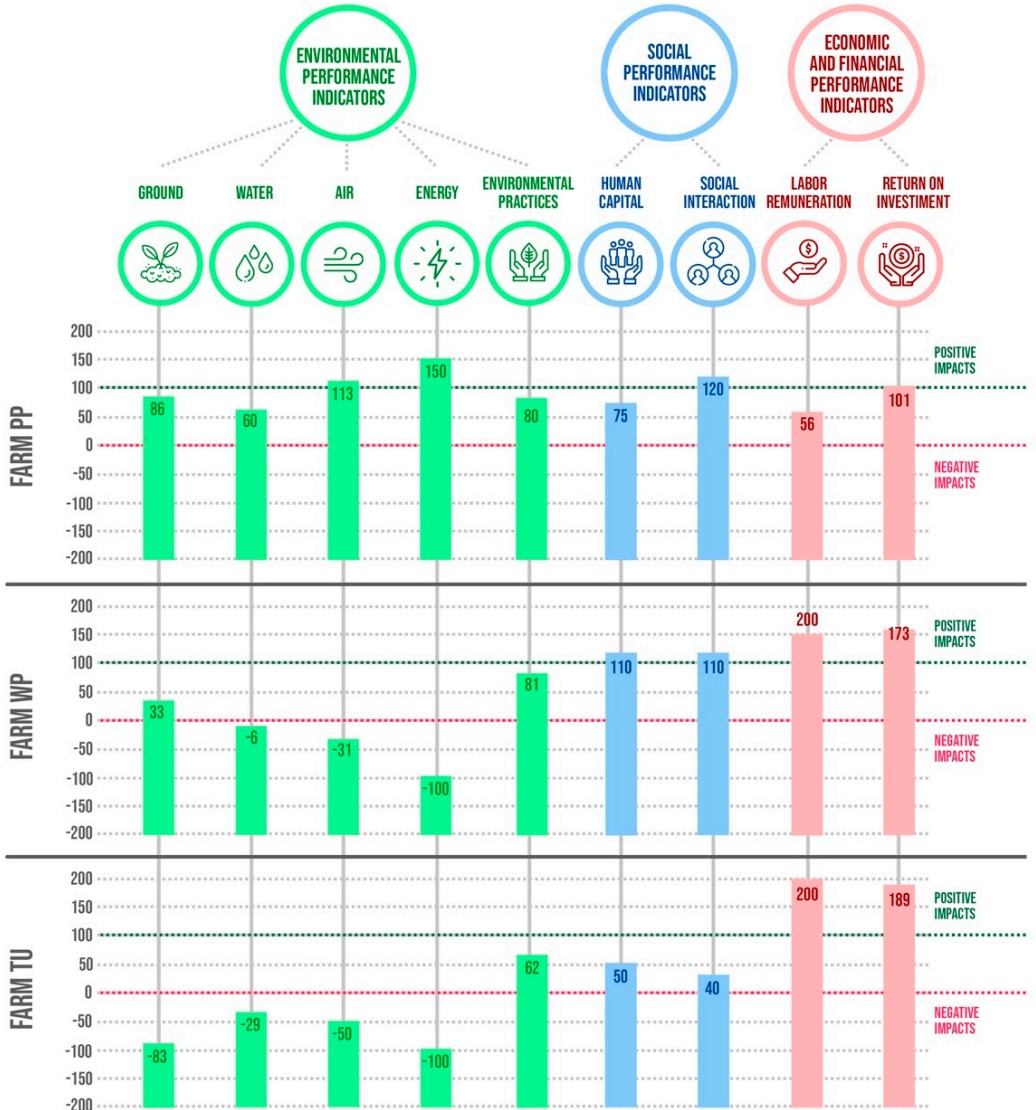

**Figure 3.** Results of the application of the model.

The level of some negative impacts in some dimensions and companies highlights the need for investment and public policies aiming, for example, towards the use of more and better treatment processes and disposal of waste solution for pig farming. Considering, for example, legislation for the use of biodigesters, which makes possible the separation of gas, solid and liquid, this liquid reused as fertilizer [15]. As also, the promotion of the use of alternatives, such as manure cages and/or composting tanks [14].

The success of these actions will also contribute positively to higher levels of family retention in rural areas. The results are similar to other studies that highlighted weaknesses and negative environmental impacts, arising from the production practices of pig farming, about negative externalities of swine production [17], TBL in livestock supply chains [1], and research on swine production carried out in China [13] and Taiwan [15].

The proposed model and the set of indicators presented here can be used to support a learning process focused on the evaluation and continuous improvement of the companies and production processes that characterize the upstream and the downstream of the swine supply chain. Furthermore, in highly integrated supply chains (as is the case in

swine production), it is essential that all players contribute to the economic, social, and environmental sustainability of the whole supply chain.

For example, previous research has shown that closed-cycle farms have advantages in terms of raising healthy animals. However, specialized breeding and finishing farms seem to be more sustainable [9]. Supply chain management in pig farming promotes the social welfare of pigs, but consumes more resources, mainly land, and has higher nitrogen leakage as a result of the higher amount of feed required to produce [34].

The TBL dimensions of swine production chains are currently insufficiently researched [49]. Farmer faces more challenges related to the improvement of their farming practices, and the continuity of small farmers is an important point when discussing the sustainability of these production systems (as well as the employment and income generation capacity of families living in rural areas) [4].

Environmental practices must be monitored and adjusted to the objectives of sustainable development right from the upstream level of the swine supply chain, composed by small farms in rural environments where the process of raising, rearing, and fattening animals takes place. There are trade-offs to be managed among TBL's dimensions and buyers-suppliers in the swine supply chain.

Table 1 shows aggregated indicators for the three sustainability dimensions and globally by farm, assigning the same weight to each metric, indicator, construct, and dimension.

**Table 1.** Farm performance by dimension and aggregate indicator.

| | Environmental Performance | Social Performance | Economic and Financial Performance | Aggregate Indicator |
|---|---|---|---|---|
| Farm PP | 98 | 98 | 79 | 91 |
| Farm WP | −5 | 110 | 187 | 97 |
| Farm TU | −40 | 45 | 195 | 66 |

The farm PP shows a good balance of the three sustainability dimensions, despite a more modest result in the economic-financial dimension. On the other hand, farms WP and TU, and particularly the latter, show a significant bias towards the economic dimension (see Figure 4).

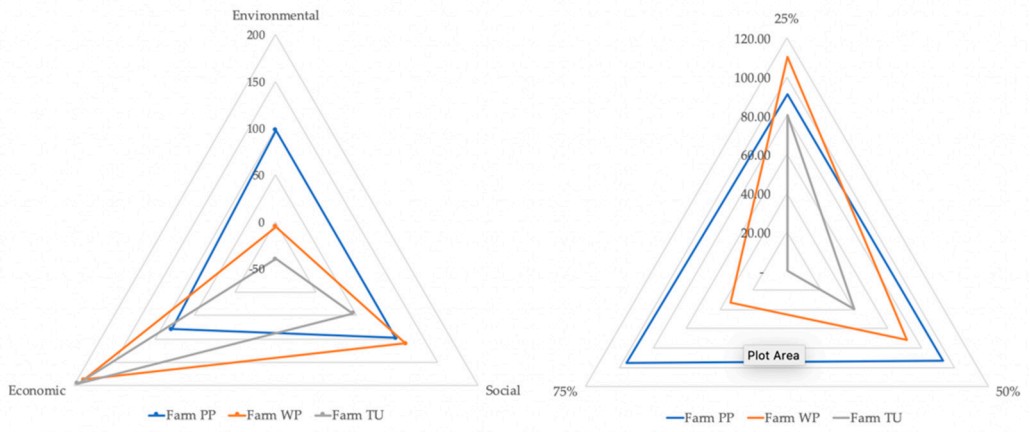

**Figure 4.** Farm performance by dimension (on the **left**) and considering different weights for TBL dimensions (on the **right**).

The results could be different if different weights than the ones established in the model are considered. The study of the results according to an admissible or possible range for the weights is important and several multicriteria techniques can be used for that purpose. For example, the graph on the right in Figure 4 simulates the aggregate indicator if the environmental dimension weights 25%, 50%, and 75% (and weighting the other two dimensions equally). It is observable that farm PP is not sensitive to these different

weights, but farms WP and TU are both very sensitive (particularly the last one). These weights depend on a series of factors (namely normative aspects related to the decision makers, contextual variables, etc.). A multicriteria-based research plan can be developed to explore and develop the weighting of the key factors of the model.

## 6. Conclusions and Opportunities for Further Research

Supply chains need to promote effective social and environmental actions to complement their economic dimension. Typically, focal companies tend to demand these actions from all of the players belonging to the supply chain. The upstream players, particularly rural producers and farmers, are important for the sustainability of the swine supply chain (namely regarding the treatment and destination of production waste).

Indeed, for the supply chain to be sustainable, it is necessary that each actor delivers satisfactory economic results, contributes to social integration, and produces without degrading the environment. Therefore, the identification and study of both positive and negative impacts of the involved companies in the swine supply chain are fundamental.

The model proposed here: (1) aggregates the three pillars of sustainability (environmental, economic and social); (2) highlights both negative and positive impacts; (3) reflects the status quo and enable the continuous assessment of the production processes; and (4) identifies bottlenecks and how to improve the production and business processes related to the swine SC.

This model differs from other previous studies (e.g., [1,4,30]), by identifying metrics and impacts at the individual company level (in this case, swine farms). The success of actions, strategies, and policies towards sustainability asks for the consideration of the specificities of each company throughout the supply chain.

The proposed model can be adapted to other supply chains, such as cattle and poultry. These models can be replicated in different locations and the different results can help to extend it including other indicators and the role of cultural and socio-economic factors. The computation of indexes can also be approached from different perspectives offering opportunities for additional work and contributions in these aspects. A continuous and comprehensive assessment of livestock supply chains is fundamental to identifying and correcting negative impacts and to understanding and replicating positive impacts, contributing to the sustainable development goals proposed by the 2030 Agenda.

**Author Contributions:** Conceptualization and methodology, S.D.K.; writing and formal analysis, S.D.K., A.Z. and P.A.; review and editing, O.D. and P.A. All authors have read and agreed to the published version of the manuscript.

**Funding:** This research received no external funding.

**Informed Consent Statement:** Not applicable.

**Conflicts of Interest:** The authors declare no conflict of interest.

## Appendix A

**Table A1.** Environmental performance indicators.

| | | | | Upper Bound | Lower Bound |
|---|---|---|---|---|---|
| | | | Physical/chemical soil analysis | | |
| Soil | - Critical Environmental Limit for Phosphorus | % | For soil evaluation, Phosphorus is one of the components that reflect the evaluation of soil quality. Its capacity of absorption and/or contamination constitutes the main parameter. Although there are differences between soils, the suggested CSL-P is 40%. It represents the environmental critical limit of Phosphorus in the soil. | 20% | 40% |
| | - Phosphorus | $Mg^3/kg$ | Represents the quantity of $mg^3/kg$ of Phosphorus present in the soil. The amount of phosphorus considered as acceptable limit is 110 $mg^3/kg$. | 60 $mg^3/kg$ | 110 $mg^3/kg$ |
| | | | Soil conservation practices | | |
| | - No-till planting | % | No-till farming is a soil conservation practice, which aims to preserve the soil avoiding the use of machines/spiders for cleaning the soil. | 90% | 70% |
| | - Crop rotation | Factors | Crop rotation contributes towards soil preservation, indicating better levels of absorption of components and nutrients, especially in the capacity to receive waste (solid or liquid). The recommendations are for temporary crops to be planted intermittently between annual crops, with some variation between yearly crops. | 3 | 2 |
| | | | Land occupation | | |
| | - Total area available/number of animals housed | $m^3/ha/year$ | Swine farms need to have a usable area available to dispose of the waste generated by production. The legislation allows third-party areas to be considered in order to compose the soil absorption capacity. As for the $m^3/hectares/year$ ratio, however, specialists recommend considering only disposal in the area itself. The reference value was 0.0091 $m^3$ of waste per sow per day, identifying the value per year and dividing by the available area. | 30 | 50 |
| | - Total available own area/waste generation | $m^3/ha/year$ | The disposition of the useful area itself and available to give disposal to the dejections generated by the production, represents an indicator of their destination and their absorption capacity of the soil (ratio $m^3/hectares/year$). However, the indication of the specialists, as the best evaluation requirement, is to consider only the disposal in the area of each rural property. | 30 | 50 |
| | - % Permanent Preservation Area | m | The Legislation of the state of Santa Catarina (Law 16.342/2014) establishes for permanent preservation areas, minimum distances between facilities/paddocks and watercourses. | 50 | 30 |

**Table A1.** *Cont.*

| | | | | Upper Bound | Lower Bound |
|---|---|---|---|---|---|
| | - | % Legal reserve | % | The current Legislation indicates the minimum of 20% of area destined as a Legal Reserve (Law 12.651/2012, Art. 12), in the context of rural properties in the Southern region. | 30% | 20% |
| Water | | | | Origin of the water source used for animal consumption | | |
| | - | Natural sources | % | For animal consumption, the following are considered natural sources: rivers, streams, springs, streams, protected sources, etc. Indicating the use of water from natural reserves. | 50 | 70 |
| | - | Artificial sources | % | Artificial sources are used for animal consumption, such as ponds, artesian wells, rainwater collection. | 40 | 20 |
| | - | Distance between installations of sources or effluents | m | The minimum distance between the sources or natural fountains is at least 30 m, as suggested by the current Environmental Legislation (Law 16.342/2014), 30 m is the shortest distance indicated for watercourses up to 10 m wide and the facilities of the swine activity. | 50 | 30 |
| | | | | Conscious use of water | | |
| | - | Use of a device to avoid wasting water | Factors | It has systems for capturing and reusing water (rainwater or process water) for the flow of waste and animal consumption. | 3 | 1 |
| | - | Reuse of water | Factors | It has systems for capturing and reusing water (rainwater or process water) for the flow of waste and animal consumption. | 3 | 1 |
| Air/ greenhouse impact | - | Greenhouse gas emissions, effluents and waste. | m$^3$ | It has a biodigester and methane is burned, minimizing the impact of greenhouse gases. | 60 | 40 |
| | | | | Air quality | | |
| | - | Community satisfaction | % | Percentage of number of days out of year which have poor air quality as a result of pig production. | 10% | 30% |
| | - | Windbreaks (dispersion of air, if there is a natural barrier). | m | Extent and width of windbreaks or natural barriers to aid air quality. Consideration is given to the extent of facilities, width of barriers (up to 5 m, between 5–10 m or over 10 m. | 2 | 0 |
| Energy | | | | Total energy use (kwh) | | |
| | - | Energy consumption | % | Rate of reduction in total energy cost and efficiency of facilities through the use of biodigesters or other technologies. | 20% | 10% |
| | - | Energy generation | % | Percentage of savings generated by improvements in conservation and efficiency of facilities, such as the use of biodigesters. | 20% | 10% |

**Table A1.** *Cont.*

| | | | | Upper Bound | Lower Bound |
|---|---|---|---|---|---|
| | | | Waste treatment | | |
| | - Process used | Factors | Identification and characteristics of the process: whether it uses a latrine, compost bin or biodigester for the treatment of waste generated by pig farming. | 3 | 1 |
| | - Destination of the waste generated by the activity | Factors | The waste from pig farming is treated and disposed of appropriately as: soil/pasture fertilizer, soil/pasture fertilizer, commercialization, or even energy generation. | 4 | 1 |
| | - Disposal of solid waste | Factors | There is proper disposal of solid waste produced in the pig farming activity: proper storage, the supplier/cooperative or the municipal government collects it through selective collection. | 3 | 0 |
| | - Disposing of dead animals | Factors | The destination or disposal of dead animals related to pig farming: whether they are composted, incinerated or frozen, and their collection. | 3 | 0 |
| | | | Environmental regularization | | |
| | - Compliance with law and regulations | Factors | The rural entity has an Environmental License, Rural Environmental Registry and participates in the Environmental Regularization Program, or even has quality programs, such as ISO 14,000 standards. | 3 | 1 |
| | - Notifications or penalties received | Factors | Forms and expenses related to environmental fines related to the activity. | 3 | 1 |
| | | | Animal welfare | | |
| Environmental practices | - Available area per head | m | By considering the ratio between the size of the facilities and the average number housed, the available area per head is identified. The larger the available area per animal, the better. | 2.50 | 2.10 |
| | - Breeder housing in collective stalls | Units | Accommodation or the housing of the breeding stock, when collective, is conducive to animal welfare. | 4 | 2 |
| | - Type of floor of the installations | Factors | The rougher it is, the more adequate for the animals (quality). Whether smooth, rough or other concrete is used. | 3 | 1 |
| | - Number of animals per water trough | Units | Adequate flow rate per production phase, considering an average of 10 to 12 animals per trough. | 10 | 12 |

**Table A2.** Social performance indicators.

| | | | | Upper Bound | Lower Bound |
|---|---|---|---|---|---|
| | Level of satisfaction with the rural environment | Score | Score given by the family regarding its satisfaction in living in a rural environment. | 9 | 7 |
| | *Work system* | | | | |
| | -    Family labour force | Persons | The labour used in the pig raising activity is family-owned, and the income from this ensures their remuneration. | 4 | 2 |
| | -    Third parties/collaborators | Factors | If third parties or collaborators are used, they are registered (regular work regime, complying with the law). | 3 | 1 |
| Human capital | *Training and development of people* | | | | |
| | -    Technical Capacity | Hours | Number of hours of annual training focused on the management of the business or pig raising activity—per member of the activity. | 20 | 10 |
| | -    Development of human capital | Factors | The technical or higher education of the managers or of the descendants who work in the activity related to the business (agronomy, rural management, etc.) is considered. | 3 | 1 |
| | Family health | Factors | Health of the managers, if there have been any illnesses which keep them away from their activities throughout the year. | 3 | 1 |
| | Family Succession | Factors | The family discusses the family succession process. The family has some sons and daughters that collaborate in the activities and are interested in the succession/ continuity of the activities. | 3 | 1 |
| Social interaction | Quality of life in the community | Groups | Social interaction, such as participation in the community (church, mothers' club, community services, etc.) or other groups in the community, is considered. Social interaction adds values of coexistence and facilitates the family's well-being in the community. The more integrated the better. | 3 | 1 |
| | Social participation | Groups | Participation in professional or rural trade unions, or in cooperative societies are factors for advice and social interaction. | 3 | 1 |
| | Social programmes | Hours | Participation of managers in education and training programs, such as workshops and training sessions to improve the quality of life in rural areas. | 15 | 5 |
| | Perception of environmental impacts | Factors | Perception of neighbours, regarding the environmental impacts of the swine activity (odor, waste, etc.). | 3 | 1 |
| | Suppliers | Factors | Considers the responsibility of input suppliers of the swine activity. Factors such as waste collection, legality of labour, instruction in use, among others. | 3 | 1 |

**Table A3.** Economic-financial performance indicators.

| | | | | | Upper Bound | Lower Bound |
|---|---|---|---|---|---|---|
| Remuneration of the workforce | | | $ | The remuneration of family labour or that of third parties must be guaranteed by the revenue obtained from commercialisation. Such minimum remuneration should be 1.5 minimum wages. In case of family labour, the net profit divided by the family members in the activity is considered. | 1.50 Minimun wage | 1 wage or less |
| ROI | - | Return on investment per pig housed | $ | The total value of investments in facilities, divided by the average number of pigs delivered per batch. This measure represents the cash generation capacity to guarantee a return on investment in a shorter time. The lower the investment value per pig, the better. | 80% of minimum wage | 20% over the minimum wage |
| | - | Net profit from activity | $ | The net profit of the activity, represents the average value of the result, considering the net revenue minus operating expenses and costs. This result is the guarantee of the return on invested capital. | 25% | 15% |
| | - | Payback time | Years | The return time of investments made in the swine activity, as well as in other enterprises, should guarantee the return of the capital invested. The return on investments considers that the shorter time, the better. Considering the need of new investments for improvements or new adaptations, 10 years are considered the longest time desired for the recovery. | 8 ys | 12 ys |

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
