# Peer review of "Performance Measurement Model for Sustainability Assessment of the Swine Supply Chain"

_sustainability, doi:10.3390/su14169926_

Round 1

Reviewer 1 Report

1. Set of experts interviewed is very less (19+2+3) al line no. 261, which needs further improvement. If not, justification for the low number may be mentioned.

2. In table A1, no. of samples collected (like to ascertain the level of phosphorus in the soil) for the study need to be mentioned as frequency.

3. There is no aggregate indicator of the model. There are different levels of expression under different categories, but for policy level, can the author arrive at one index of aggregate value.

4. Innovativeness of the model is not uttered out, so the authors should indicate much innovativeness in the study rather than an elaborated introduction.

5. Introduction and conceptual framework are very elaborative where readers may be lost. Try to make it simple and small.

6. Self-citation of Zanin is found in many places (ref no. 43, 47, 49, 50, 51, 52, 61) which should be made to a minimum.

Author Response

Dear editor and reviewers,
Many thanks for all valuable comments and the opportunity to improve our paper.
The changes made are highlighted in the revised version and below we explain how the comments and suggestions made by the reviewers were addressed.
Best regards.
The authors.

Author Response

(The authors gave the same response as above.)

Round 2

Reviewer 2 Report

the article is fine, the authors have complied with all remarks and the article can be accepted in the present form.

This manuscript is a resubmission of an earlier submission. The following is a list of the peer review reports and author responses from that submission.

Round 1

Reviewer 1 Report

The paper “Performance measurement model for cost-benefit analysis and sustainability assessment of livestock supply chains ” starts with an intersting aims of evaluating swine farms with a multimensional approach, but at the end in the main findings this multidimensional spirit is not considered while instead of considering the results in unique way they are used as separate indicators.

Anyway the reading of the paper is complex and difficult to follow what the authors want to transmitt to the reader. Moreover, the text in the manuscript appear a bit messy with some parts of text not in the proper area of discussion (for example the comments on the methods are really scarce, whereas some methodological indications are provided in the results section).

The authors spent a lot of time to introduce the topic of sustainability taking it from a very large point without really focusing on the swine industry.

I suggest to carefully review all the paper starting from its own structure and then shortening the text in which many redundant sentences and concepts are present. Many sentences should be rephrased to give more sense to what the authors want to transmitt to the reader, I suggest an english revision focusing not on the grammar, but more on the story that the paper tells.

The title also is very measleading, the authors used the term “Cost-Benefit Analysis” but they neither comment this in the whole manuscript nor employ the CBA as a method of analysis. What they perform is not at all a CBA that has a strong economic base with an impressive (old and new) literature.

In some part of the manuscript the information provided are not correct:

“In this context, several countries have signed a series of international agreements regarding the maintenance of the environment (Kyoto Proto, Montreal Protocol, etc.), with the purpose of protecting the planet's genetic diversity

Kyoto and Montreal Protocol do not refer to genetic diversity, but overall impacts of GHG and CFC emissions.

In other parts the text is not well supported by references:

sustainability assessment of production systems and supply chains, the environmental component has prevailed.

This is not totally true, there is plenty of papers which consider sustainability in agriculture employing multidimensional approaches (e.g. Multicriteria analysis)

In section 2.3 livestock supply chain the environmental issue related to agriculture are only mentioned at the end of the section:

However, the agricultural sector increasingly uses non-renewable natural resources to meet the needs and requirements of the market. Moreover, its expansion and evolution cause significant environmental impacts, without taking into account the various dimensions of sustainability.

The authors miss to discuss in this section the great impacts of the agricultural sector on the environment in terms of pollution, soil degradation, erosion, biodiversity loss, deforestation etc..

Doing this they also miss the opportunity to highlight the importance of their study. I suggest to add at least two-three paragraphs in which you comment those issues using scientific references and/or international agency reports.

The case studies as the area of study should be discussed in a separete section. Now the swine sector in Santa-Caterina state is commented in the results and that should not be there.

The discussion of the method is overall unclear and it should be substantially improved, because as it is now it is very difficult to understand how your analysis was built, what you did in empirical terms and therefore the overall contribution of your study.

Tables are not in line and some pictures, even if they are nice, are not very explicative (Fig. 1 and 2).

Moreover, I would suggest to separate the results section from the disussion (now that part is very confusing). In addition, the conclusion should be strongly reduced while all the comments, policy advice and limitations should go in the discussion part. In the concluding remarks the authors may comments the main contribution of their study and briefly summarize what they did (max half page). In attachment the paper with some minor comments.

Reviewer 2 Report

Here are a few suggestions for making your text even better.

The objective was no clearly defined, in lines 346-347 it is stated “the purpose of a performance measurement and evaluation model for cost-benefit analysis and sustainability assessment of livestock supply chains”, and in lines 13-14 “a model and a set of indicators for evaluating the sustainability of the swine production in the Southern region of Brazil”. It is suggested that the authors precisely define the object of study in the research, whether they are evaluating the sustainability of the swine production or livestock supply chains. It is important to specify or justify each terminology used in their manuscript (livestock supply chains, sustainability of the swine production, sustainability assessment of livestock supply chains, companies in the swine supply chain, livestock supply chains, agro-industrial production, agro-industrial supply chains, sustainability of activities developed in rural areas, the agricultural sector, the agricultural system, development of sustainable business, food industry, agri-food supply chain, sustainable supply chain, family farming systems, supply chain).

The authors need to revise their manuscript in the light of the most recent literature regarding sustainability of the swine production instead of livestock supply chains.

In the methodology the authors must state:

a)    Model programming and parameterisation, and a description of the specific assumption considered for the development of sustainability assessment model for the swine production chain (line 355)

b)    Software systems for model development or validation,

c)     structure of the model

d)    guidelines followed to evaluate human capital and social interaction; economic and financial performance, and environmental impact.

The conclusion does not summarize the findings; it does not state implications of the results.

The content of the various sections of the manuscript (Results and Discussion) was appropriate. Please rethink Abstract, Introduction, Materials and Methods, and Conclusions

Specific comments below.

Line 66: Another related issue is the concept of circular economy…. Please, establish the connection with the last paragraph

Lines 74-75: The extant literature and current research have been offering relevant contributions both theoretical and practical on these topics ..Which topics?
Line 78:
sustainability and CBA perspectives in …. Establish the connection with the last paragraph

Line 337: Twenty-four specialists. What type of specialists.

Line 342:  the 24 experts. What type of experts.

Line 620: rural producers and farmers are important for the sustainability of the swine supply chain….. It is not in the results.

Lines 626-627:  The southern region of Brazil holds 66.5% of the Brazilian pig production, according to 626 the Brazilian Institute of Geography and Statistics (2022)…This is not a conclusion

Lines 661: As a way of testing the model…… this sentence in the conclusion, changes the purpose of the manuscript

Reviewer 3 Report

Dear Authors,

I think this paper has potential to contribute to Sustainability.  It is a well thought-out and interesting study but there are some issues to reconsider. Below I have some comments for the Authors to address:

Introduction

L 100 2030 Agenda ADD “for Sustainable Development”

L 114-123 The aims of the study should be presented  at the end of the introduction.

L136 should be:  livestock supply chain

L162 Why you do not quote Paris agreement instead of Kyoto and Montreal?

L 303-304 for consistency add SDG

L 311 Present more in depth the link between animal welfare and sustainability  

Material and Methods

Table 1 – the information on the selection of the initial set of indicators should be provided in a comprehensive manner

L 354 The limitation related to the validation process should be clearly highlighted as regards the number and type of farms. The study is of rather preliminary nature.

L 372 Provide more details on the interview process

L441 The units should be described more in detail.

L 442 Such piglets? Which exactly?

L 465 the need to improve animal welfare is not well placed in the context of this paragraph. Discuss it more in depth.

L 486 How the studies in Brazil you mention relates to your study?

L489 sounds confusing - Do you mean previous study or your study?

L 495 employment and income generation are not included in the figure 3.

Subchapter 4.3. Reflect the issues arising from the limited number of farms clearly in the limitation of the study.

Reviewer 4 Report

It is possible to make the introduction shorter. It appears to me that a great deal of irrelevant information has been included. It needs to be rewritten with a clear focus.

Studies that are not directly relevant are also included in literature reviews. It must be written with a clear focus.

It is necessary to explain the data. For example, the Delphi method, the initial sample, the final results, dropout, the questionnaire details, the respondents' backgrounds, and so on.

The conclusion can be improved with a clear focus